

# Online verification and management scheme of gateway meter flow in the power system by machine learning

Chong Li[1], Hao Wang[1], Hongtao Shen[1], Peng Yang[2], Yi Wang[1], Qian Li[1], Chuan Li[1], Bing Li[1], Rongkun Guo[1] and Ruiming Wang[1]

[1] State Grid Hebei Marketing Service Center, Shijiazhuang, HeBei, China
[2] State Grid Hebei Electric Power Company, Shijiazhuang, HeBei, China

## ABSTRACT

Currently, the calibration of electric energy meters often involves manual meter reading, dismantling inspection, or regular sampling inspection conducted by professionals. To improve work efficiency and verification accuracy, this research integrates machine learning into the scheme of online verification and management of gateway meter flow in the power system. The approach begins by applying the Faster Region Convolutional Neural Network (Faster-RCNN) model and the Single Shot MultiBox Detector (SSD) model to the recognition system for dial readings. Then, the collected measurement data is pre-processed, excluding data collected under light load conditions. Next, an estimation error model and a solution equation for the electricity meter are established based on the pre-processed data. The operation error of the electricity meter is estimated, and the estimation accuracy is verified using the limited memory recursive least squares algorithm (LMRLSA). Furthermore, business assistant decision-making is carried out by combining the remote verification results with the estimation outcomes. The proposed dial reading recognition system is tested using 528 images of meter readings, achieving an accuracy of 98.49%. In addition, the influence of various parameters on the error results of the electricity meter is also explored. The results demonstrate that a memory length ranging from 600 to 1,200 and a line loss error of less than 5% yield the most suitable accuracy for estimating the electricity meter error. Meanwhile, it is advisable to remove measurement data collected under light load to avoid unnecessary checks. The experiments manifest that the proposed algorithm can properly eliminate the influence of old measurement data on the error parameter estimation, thereby enhancing the accuracy of the estimation. The adjustment of the memory length ensures real-time performance in estimating meter errors and enables online monitoring. This research has certain reference significance for achieving the online verification and management of gateway meter flow in the power system.

Corresponding author
Chong Li, chunglee3181@126.com

## INTRODUCTION

Currently, power companies primarily verify the accuracy of the gateway meter flow (referred to as the total meter) in the power system through dismantling inspection or periodic sampling inspection conducted by professionals (*Liu et al., 2021*). However, with the vast power grid in China, comprising over 500 million meters, the existing verification approach faces challenges in terms of high work, lengthy verification periods, low management efficiency, and the inability to meet the requirements of smart meter maintenance and replacement (*Dai et al., 2020*). To transition from regular verification to state verification of smart meters and ensure measurement accuracy, it is imperative to explore an efficient and precise online remote verification and management solution for the operation of the total meter. The traditional safety analysis method for online check of the gateway meter flow in the power system is the point-by-point method. This approach involves performing power flow calculation simulations for each potential operating mode of the system and analyzing the simulation results to identify stable safety risks in different operating modes. To reduce computational complexity, behavioral indicators can sort expected accidents, and the point-by-point simulation is performed accordingly until no new high-risk accidents are found. However, as the power grid's operating modes become increasingly diverse, the traditional point-by-point method struggles to meet the demands of power grid security analysis in this evolving context. The existing online remote verification methods for smart meters, based on measurement data analysis, primarily include ordinary least squares inversion and weighted recursive least squares. However, these methods suffer from low solution accuracy and practicability, and they are susceptible to factors such as power consumption levels, the number of user meters, and data quality (*Liu, Liang & He, 2020*). *Wang et al. (2020)* proposed an error analysis approach for electricity meters based on advanced meter infrastructure measurement data. By comparing existing meters within a cluster, they calculated the error without the need for external standard instruments. *Wang et al. (2019)* proposed a machine learning (ML)-based power system attack detection model. The model utilized information and logs collected by phasor measurement units, with a random forest chosen as the basic classifier of AdaBoost. The experimental results demonstrated an accuracy rate of 93.91% and a detection rate of 93.6%. *Zhang et al. (2022)* provided an overview of the basic principle, research progress, training methods, typical structure, and application characteristics of deep learning (DL). They summarized the application status of DL in frequency situation awareness, frequency security and stability assessment, frequency regulation, and other aspects. They also discussed the adaptability of DL applications to various problems. Lastly, the development trend of DL and its application in power system frequency were prospected. *Guo et al. (2022)* introduced a real-time dynamic optimal energy management based on a deep reinforcement learning algorithm. They employed proximal policy optimization to capture the uncertain characteristics of renewable energy generation and load consumption through historical data. The effectiveness and computational efficiency of the proposed method were verified with an example. However, this method requires consideration of algorithmic factors, such as the number of measurement periods, the

number of meters in the station area, and the out-of-tolerance of a single low-voltage user meter on the model results. The Marketing Department of State Grid Corporation of China has researched the state inspection plan for electricity meters. The operation state of the meters is scored through four triggering methods: family defects, online monitoring, on-site inspection, and regular triggering. Corresponding inspection strategies are generated according to the scoring results (*Tan et al., 2020*). However, the analysis results provided by this research qualitatively divide the electricity meter into several states, and the analysis results are relatively coarse. Thus, it is not possible to achieve accurate remote verification of the electricity meter.

An online verification and management scheme of gateway meter flow in the power system by ML is proposed to simplify the verification method of the total meter and improve the verification accuracy. Firstly, the accurate automatic meter reading is realized through the implementation of a dial reading recognition system, incorporating the Faster Region Convolutional Neural Networks (Faster-RCNN) model and the Single Shot MultiBox Detector (SSD) model. Secondly, the data obtained from automatic meter readings are used to estimate the operation error of the electricity meter. The limited memory recursive least squares algorithm (LMRLSA) is employed for precise estimation. The error estimation results of the electric meter are then verified and analyzed. Subsequently, the online remote verification results are combined with the business decision-making process. Finally, the proposed scheme undergoes rigorous verification and testing to validate its scientific validity and feasibility. The research offers significant insights for realizing remote intelligent online meter reading and verification management of electricity meters.

# DESIGN OF THE SCHEME

## Faster-RCNN model and SSD model

The Faster-RCNN model, first proposed in 2015, is characterized by its large size and strong feature extraction capability. Due to these qualities, it is well-used for accurately identifying the specific readings in dial images. However, it should be noted that its running speed is relatively slow (*Mansour et al., 2021*). The network structure of the model is presented in Fig. 1.

In the proposed approach, the input image is initially resized to a fixed size. Subsequently, the feature map is extracted using the convolutional neural network. The region proposal networks (RPN) then generate candidate boxes based on the feature map. These candidate boxes, along with the feature map, are passed through the Regions of Interest pooling layer to obtain a fixed-size proposal feature map. Finally, the candidate boxes are classified and refined through a fully connected layer to get a more accurate target frame (*Lv et al., 2020*; *Li & Zhou, 2020*).

The SSD model is renowned for its compact size, exceptional accuracy in detecting large targets, and its ability to perform fast detection (*Yang et al., 2021*). Thus, the SSD model is employed to detect the effective reading area of the dial image. The core of the SSD model involves predicting the category score and offset of the candidate box. Simultaneously, it can generate candidate boxes of different sizes by predicting feature maps at different scales (*Zhang et al., 2021a*). The network structure of the SSD model is displayed in Fig. 2.

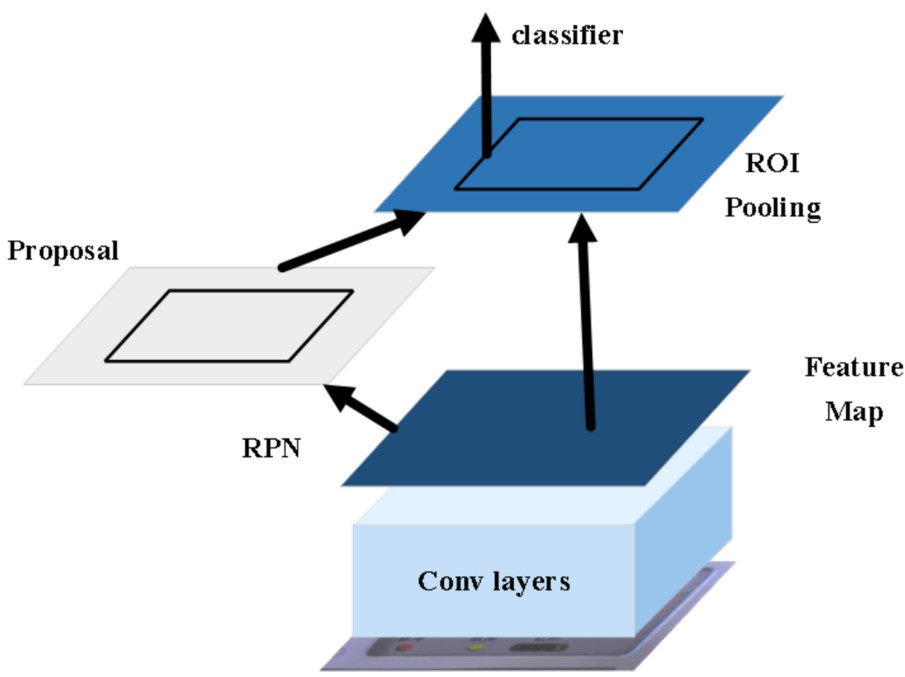

**Figure 1** **The network structure of the Faster-RCNN model.** The Faster-RCNN model is used to identify the specific readings of the dial image, but its running speed is relatively slow.

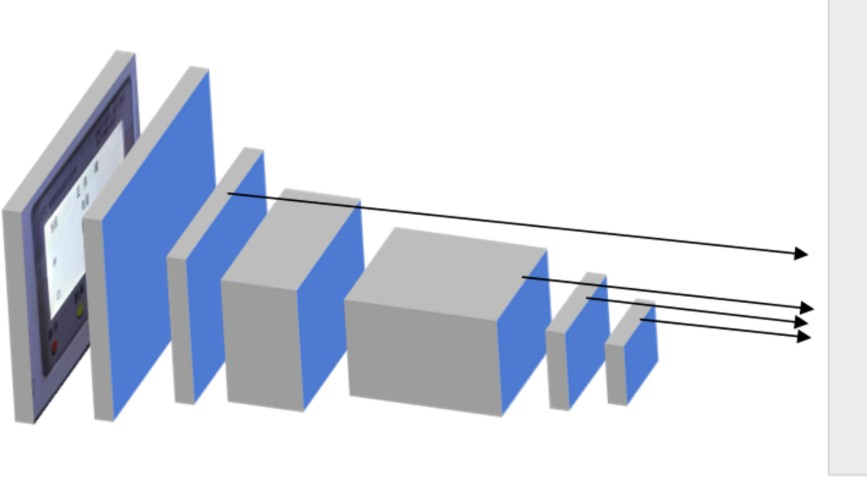

**Figure 2** **The network structure of the SSD model.** The core of the SSD model is to predict the category score and offset of the candidate box. Simultaneously, candidate boxes of different sizes can be obtained by predicting on feature maps of different scales.

The SSD model differs from the Faster-RCNN model in terms of object detection. The Faster-RCNN model initially generates candidate regions through RPN, followed by classification of these candidate regions and followed by classification of location information. In contrast, the SSD model obtains a series of candidate regions on feature maps of various scales. Each point on these feature maps corresponds to different positions of the original image, thereby enabling high-precision detection results (*Zhang et al., 2021b*; *Sindhwani et al., 2021*). When limited computational resources or faster speeds are required, alternative algorithms can be employed for object detection tasks. For instance, You Only Look Once (YOLO) and RetinaNet are viable options. YOLO excels in speed, facilitating real-time detection in applications that require swift processing. On the other hand, RetinaNet utilizes a feature pyramid network structure, enabling inspections at multiple scales. RetinaNet can handle targets of different sizes and maintain high accuracy efficiently by conducting inspections at multiple scales.

While the SSD model boasts superior speed, the Faster R-CNN model typically outperforms it in terms of target detection accuracy. The Faster R-CNN model incorporates a Region Proposal Network that generates candidate regions, subsequently subjecting them to classification and positioning. This two-stage design empowers the Faster R-CNN to achieve greater accuracy when confronted with intricate scenes and small-scale targets.

## Realization of image recognition function

This research utilizes the Google TensorFlow framework for model training. As an excellent development framework for ML, TensorFlow provides developers with various mature model implementation solutions. In particular, TensorFlow provides the TensorFlow Object Detection API, a part of the TensorFlow models subproject, which is employed for object detection tasks. This framework is renowned for its extensive application in Google's computer vision projects. Leveraging an open-source framework, it facilitates the easy construction, training, and deployment of object detection models (*Rani et al., 2021*; *Ghifari, Darlis & Hartaman, 2021*; *Sajeevan et al., 2020*). For electricity meter reading recognition, two models are mainly trained. The first model focuses on training the SSD model to detect the effective reading area on the electricity meter screen. The second model centers on training the Faster-RCNN model to identify the readings within the detected effective reading area. Figure 3 illustrates the specific training process of the SSD model.

During the training of the SSD model, several steps are followed. First, the input meter image is resized to ensure consistency. The height of the image is uniformly set to 500 pixels, while the width is adjusted proportionally to maintain the original aspect ratio. To define the effective reading area of the image screen, the LabelImg software is used for manual annotation. This process creates a training sample set and generates an XML format file containing the annotated information. All labeled data is utilized to construct the training sample set and generate the corresponding XML format files. Next, the generated XML format files and their corresponding images are converted into TFRecord format files. Finally, the pre-trained model is loaded, and the TFRecord format files are used for model training. The desired SSD model is obtained through this process (*Shi et al., 2020*; *Wang et al., 2021*).

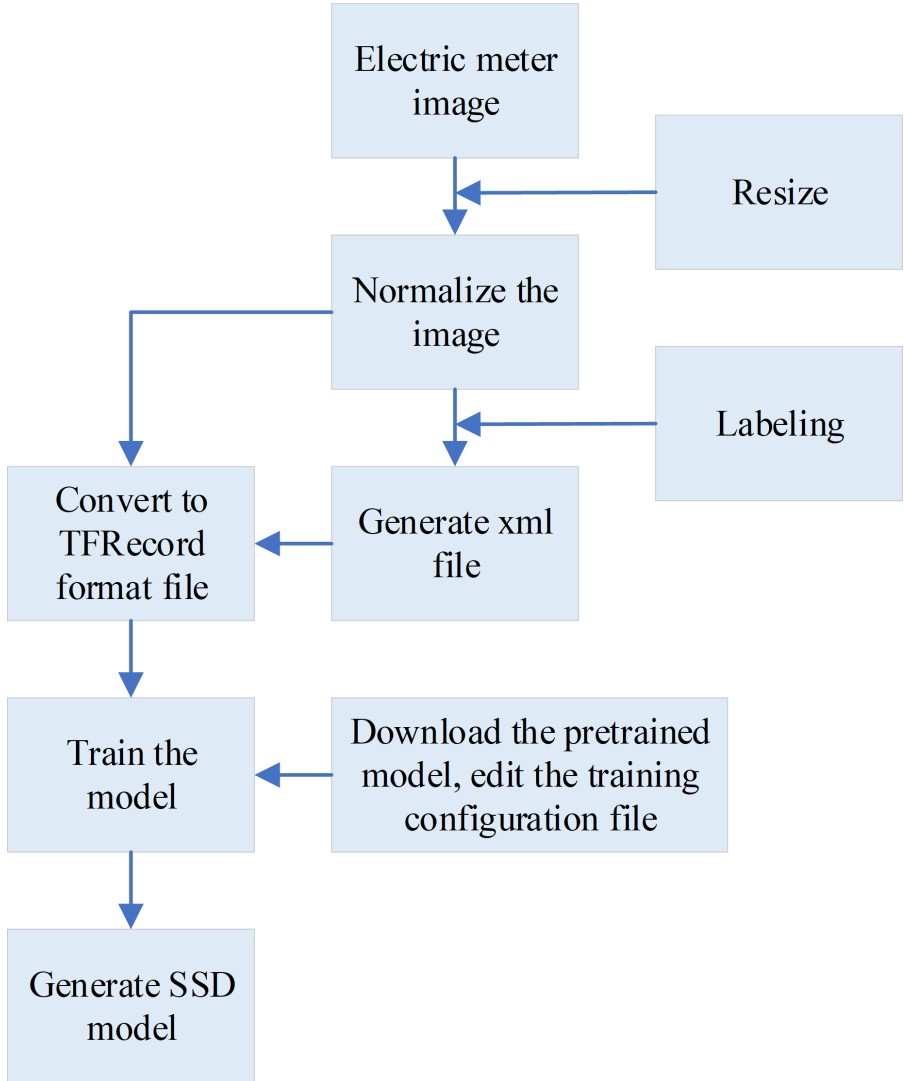

**Figure 3** **The specific training process of the SSD model** (*Song & Wen, 2020*)**.** When training the SSD model, first, the input meter image is resized, the height of the image is uniformly set to 500px, and the width is scaled proportionally. Then, LabelImg software is used to label the effective reading area of the image screen, a training sample set is constructed, and an XML format file is generated. Next, the generated XML format files and images are converted into TFRecord format files. Finally, the pre-trained model is read and the TFRecord format file is used for model training, resulting in the desired SSD model.

The training process of the Faster-RCNN model is plotted in Fig. 4.

To train the Faster-RCNN model for identifying readings within the effective reading area, the following steps are followed. Firstly, the effective reading area is processed in grayscale. Besides, the brightness is adjusted, which can solve the problem that some images are too bright or too dark and reduce the complexity of identifying the readings in the effective reading area. Second, LabelImg software is also utilized to label the numbers in the area, generating an xml format file. This file is then converted into a TFRecord format file, suitable for training the model. Lastly, the model is trained to obtain the desired

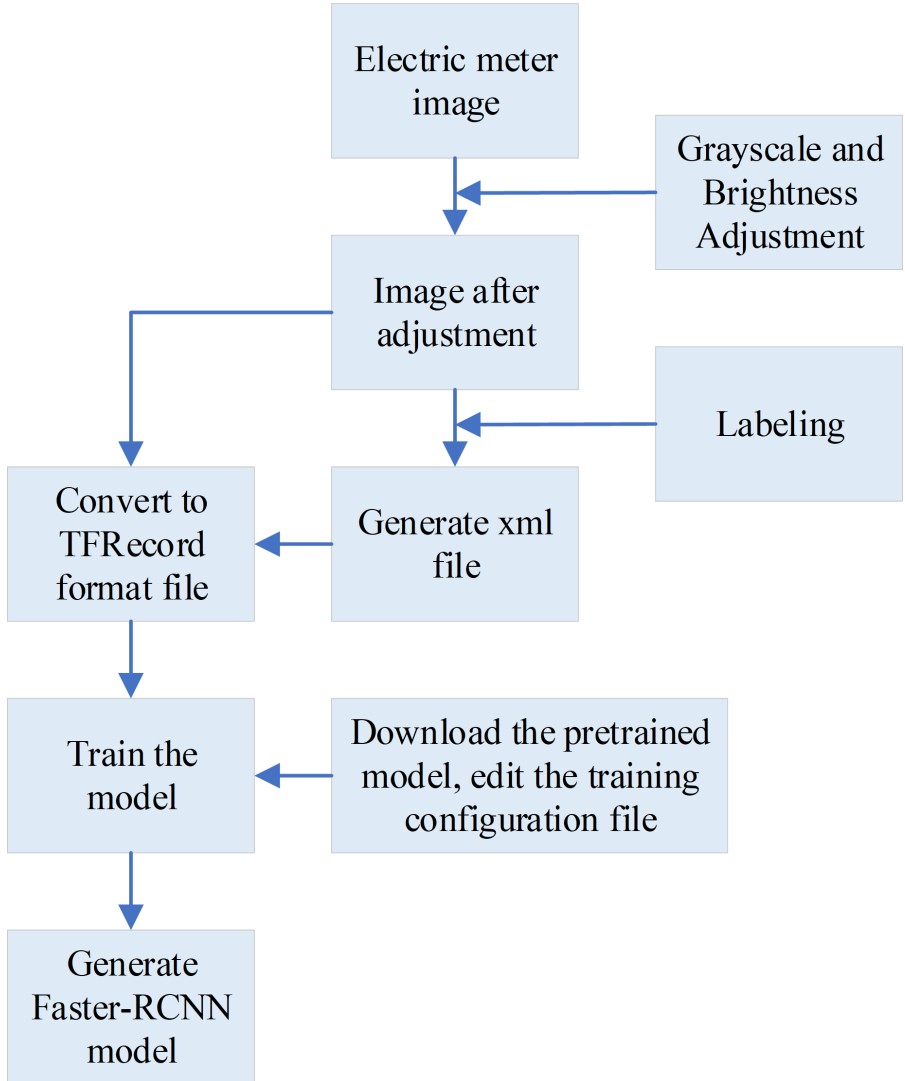

**Figure 4  The training process of the Faster-RCNN model.** For the Faster-RCNN model trained to identify the readings in the effective reading area, the effective reading area is firstly processed in grayscale, and then the brightness is adjusted, which can solve the problem that some images are too bright or too dark, and reduce the complexity of identifying the readings in the effective reading area. Secondly, LabelImg software is also used to label the numbers in the area, generate an xml format file, and convert it into a TFRecord format file. Finally, the model is trained to obtain the desired Faster-RCNN model.

Faster-RCNN model (*Xiong et al., 2021*; *Albahli et al., 2021*; *Li et al., 2020*). The specific training steps for the two models are shown in Fig. 5.

## Analysis of measurement data

The Faster-RCNN and the SSD models are employed to gather electricity metering information from both the total meter and sub-meters at predefined intervals. The collected data is then transmitted to the master station, where it is automatically copied and aggregated. Pre-processing techniques are applied to the gathered measurement

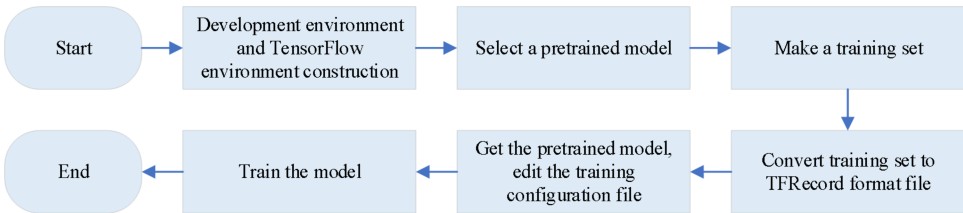

**Figure 5** **The training steps for the models.** The specific training steps for the two models.

data. Subsequently, an estimation model and solution method are established through the pre-processed data. The estimation accuracy is evaluated to accomplish the goal of online verification and assisting in business decision-making (*Cui et al., 2021*; *Kalinov & Rimlyand, 2020*). The specific process is illustrated in Fig. 6.

The improved fuzzy C-means clustering technology pre-processes the time series of original measurement data. This pre-processing step involves excluding the measurement data under light load conditions, which serves as an input variable for the online remote verification model (*Zhao et al., 2021*). The specific steps for pre-processing time series of the original measurement data based on the improved fuzzy C-means clustering are outlined below:

Step 1: The weighting index $w$ and the iteration termination parameter $\sum$ are determined according to the ratio of the increment of the electricity meter reading and the range of the electricity meter, ensuring that the difference is less than 0.1.

Step 2: The number of clusters and cluster centers are determined. The measurement data samples are sequentially selected and substituted into the hill-climbing function, as expressed in Eq. (1):

$$\hat{M}^1(x_r) = \sum_{j=1}^{n} e^{-\alpha \|x_j - x_r\|^2} \tag{1}$$

In Eq. (1), $x_j$ refers to the $j$th sample; $n$ indicates the total number of samples; $x_r$ means the $r$-th sample, which is the cluster center. $\hat{M}^1(x_r)$ signifies the hill-climbing function when the $r$-th sample is taken as the cluster center, $\alpha$ stands for a positive number. If $x_r = x_1^*$, where $x_1^*$ is a sample in the sample set, get the first hill-climbing function to get the maximum value $\hat{M}_{max}^1 = \max(\hat{M}^1(x_1^*))$, $x_1^*$ can be taken as the first cluster center (*Kumar, Sharma & Sharma, 2021*). When searching for other cluster centers, to eliminate the influence of $x_1^*$, the revised $t$-th hill-climbing function can be written as Eq. (2):

$$\hat{M}^t(x_r) = \hat{M}^{t-1}(x_r) - \hat{M}_{max}^{t-1} \sum_{j=1}^{n} e^{-\beta \|x_j - x_{t-1}\|^2} \tag{2}$$

In Eq. (2), $\hat{M}^t(x_r)$ means the updated hill-climbing function; $\hat{M}^{t-1}(x_r)$ refers to the hill-climbing function of the previous step; $\hat{M}_{max}^{t-1}$ represents the maximum value of the hill-climbing function of the previous step. The process of identifying cluster centers continues until the convergence condition $\hat{M}_{max}^t / \hat{M}_{max}^1 \leq \delta$ is satisfied, where $\delta$ is the

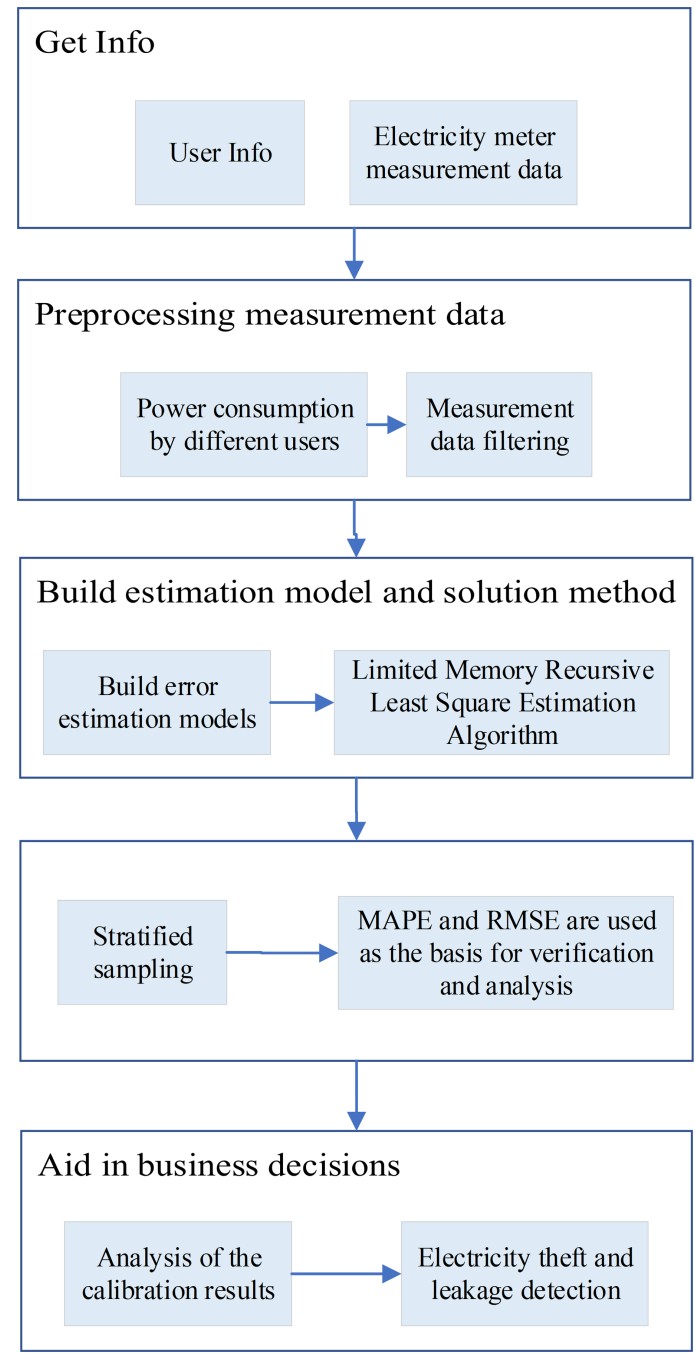

**Figure 6** **The process of scheme.** The Faster-RCNN model and the SSD model are used to collect the electricity metering information of the total meter and sub-meters according to the pre-set time, and after the collection and summary, the user's electricity consumption information is transmitted to the master station, and the electricity consumption data is automatically copied and collected. The obtained measurement data is preprocessed, and then the estimation model and solution method are established through the preprocessed data. The estimation accuracy is checked, and finally, the purpose of online verification and business assistant decision-making is realized.

convergence coefficient of the classification, typically set to 0.001 (*Nasir, Samsudin & Shabri, 2020*). The total number of iterative clustering before convergence is determined as the classification number $c$ in fuzzy clustering. At each clustering, the sample with the largest value of the hill-climbing function is denoted as $x_t^*$.

Step 3: The membership matrix is calculated using Eq. (3).

$$
\begin{cases}
x_r = \dfrac{\sum_{j=1}^{n} (u_{rj})^w x_j}{\sum_{j=1}^{n} (u_{rj})^w} \, (1 \le r \le c) \\[4mm]
u_{rj} = \left[ \sum_{k=1}^{c} \left( \dfrac{\|x_j - x_r\|}{\|x_j - x_k\|} \right)^{2/(w-1)} \right]^{-1} (1 \le r \le c, 1 \le j \le n)
\end{cases}
\tag{3}
$$

In Eq. (3), $u_{rj}$ represents the membership degree of the $j$th sample with respect to the $r$-th cluster center. The weighting index $w$ signifies determines the fuzziness of the final clustering effect and has a value range of $[1, +\infty)$. For this method, a value of 1.8 is chosen to achieve the desired clustering outcome.

Step 4: The objective function is calculated as the weighted sum of squares of distances from each sample to all cluster centers, given by Eq. (4).

$$
J_w(\mathbf{U}, \mathbf{V}) = \sum_{j=1}^{n} \sum_{r=1}^{c} u_{rj}^w \|x_j - x_r\|^2
\tag{4}
$$

Equation (4) serves as the iterative equation. The clustering process terminates when the two iteration errors $\Delta J_w(\mathbf{U}, \mathbf{V})$ before and after the objective function are less than the termination parameter $\sum$. By eliminating the light load condition from the time series of the original measurement data, the data pre-processing stage is completed.

After the data is pre-processed, an equation is established for solving the meter error. The remaining measurement data is arranged in chronological order to create the total and sub-meter matrices, respectively, which serves as input samples for the error solution of the LMRLSA (*Yadav, Mohan & Yadav, 2020*). Based on the law of energy conservation, the reading of the total meter in any measurement period is equal to the sum of the true values of each user sub-meter plus the sum of the line losses in this period (*Bohnker & Breuer, 2020*). For the $t$-th measurement period, the relationship between the total meter and the sub-meter readings can be expressed as Eq. (5).

$$
y_0(t) = \sum_{i=1}^{m} z_i(t)(1 + \xi_i(t)) + w_{\text{loss}}(t)
\tag{5}
$$

In Eq. (5), $y_0(t)$ refers to the reading increment of the total meter in the measurement period; $z_i(t)$ and $\xi_i(t)$ indicate the reading increment and error of the $i$th sub-meter in the measurement period; $z_i(t)(1 + \xi_i(t))$ means the actual power value consumed by the $i$th sub-meter in the measurement period; $w_{\text{loss}}(t)$ stands for the power consumption of all lines in the measurement period; $m$ signifies the total number of all sub-meters. The Levenberg–Marquardt (LM) algorithm optimizes the multi-layer feedforward neural network model to calculate the line loss (*Mulashani et al., 2021*). Then, $1 + \xi_i(t)$ in Eq. (5) is represented by $\theta_i(t)$, and a set of measurement data series is formed by each unit

measurement period $z_i(t)$ and the solution obtained $y(t)$. Equation (6) is derived by expressing Eq. (5) in matrix form.

$$\begin{cases} y(t) = \boldsymbol{Z}(t)\hat{\Theta}(t) \\ \hat{\zeta}(t) = \hat{\Theta}(t) - \boldsymbol{I} \end{cases} \tag{6}$$

In Eq. (6), $\boldsymbol{Z}(t) = [z_1(t), z_2(t), \ldots, z_m(t)]$ represents the measurement data matrix of each user's sub-meter in $t$ th periods; $\hat{\Theta}(t) = [\theta_1(t), \theta_2(t), \ldots, \theta_m(t)]^T$ refers to the error parameter matrix to be estimated by each user's sub-meter in the $t$-th measurement period; $\theta_i(t)$ denotes the operating error parameter of the $i$th meter to be found in the $t$-th measurement period. $\hat{\zeta}(t) = [\hat{\zeta}_1(t), \hat{\zeta}_2(t), \ldots, \hat{\zeta}_m(t)]^T$ is the remote estimated value of the operating error of the electricity meter in the $t$th measurement period. Once the equation is established, the operation error of the electricity meter is estimated through the LMRLSA. The specific steps are as follows:

Step 1: The initial value $\hat{\Theta}(0,0)$ and $\boldsymbol{P}(0,0) = \alpha \boldsymbol{I}$ are selected, where each element of $\hat{\Theta}(0,0)$ is 0 or a small number, $\boldsymbol{P}(0,0) = \alpha \boldsymbol{I}$, where $\alpha$ is a sufficiently large positive number, typically ranging from $10^5$ to $10^{10}$, and $\boldsymbol{I}$ is the unit matrix. $L$ represents the memory length, and $T$ refers to the latest measurement period. Compare T and L. If T $\leq$ L, proceed to Step 2; if T>L, proceed to Step 3.

Step 2: When T $\leq$ L, the ordinary Recursive Least Squares algorithm is used to obtain the initial parameter estimate $\hat{\Theta}(0, L-1)$, the corresponding $\boldsymbol{P}(0, L-1)$ and the gain matrix $\boldsymbol{K}(0, L-1)$, and use it as the initial quantity of the LMRLSA. The specific process is as follows:

a. After the previous $t$ measurements, the matrix equation can be obtained as illustrated in Eq. (7).

$$\hat{\Theta}(t) = \left(\boldsymbol{Z}(t)^T \boldsymbol{Z}(t)\right)^{-1} \boldsymbol{Z}(t)^T \boldsymbol{Y}(t) \tag{7}$$

In Eq. (7), $\boldsymbol{Z}(t) = \begin{bmatrix} z_1(1) & z_2(1) & \cdots & z_m(1) \\ z_1(2) & z_2(2) & \cdots & z_m(2) \\ \vdots & \vdots & \ddots & \vdots \\ z_1(t) & z_2(t) & \cdots & z_m(t) \end{bmatrix}$ is the sub-meter matrix, and $\boldsymbol{Y}(t) = \begin{bmatrix} y(1) \\ y(2) \\ \vdots \\ y(t) \end{bmatrix}$ is the total meter matrix. It is assumed that $\hat{\Theta}(t-1)$ has been calculated in the $(t-1)$st recursion. Before the $t$-th recursion, the time series of the new total and sub-meter measurement data collected are $y(t)$ and $\boldsymbol{Z}(t) = [z_1(t), z_2(t), \ldots, z_m(t)]$. All the measurement data of each user sub-meter measured in the previous $t$ times are represented by $\boldsymbol{Z}_t$, and all the measurement data of the total meter measured in the previous t times are represented by $\boldsymbol{Y}_t$. The previous $t-1$ measurements are represented by $\boldsymbol{Z}_{t-1}$ and $\boldsymbol{Y}_{t-1}$, respectively. Based on the measurement data collected in the previous $(t-1)$st measurement and the previous $t$-th measurement, the estimation result of the meter error parameter is given by Eq. (8).

$$\begin{cases} \hat{\Theta}(t-1) = \left(\boldsymbol{Z}_{t-1}^T \boldsymbol{Z}_{t-1}\right)^{-1} \boldsymbol{Z}_{t-1}^T \boldsymbol{Y}_{t-1} \\ \hat{\Theta}(t) = \left(\boldsymbol{Z}_t^T \boldsymbol{Z}_t\right)^{-1} \boldsymbol{Z}_t^T \boldsymbol{Y}_t \end{cases} \tag{8}$$

b. The inverse $P(t)$ of the covariance matrix of the measurement data is calculated according to Eq. (9).

$$
\begin{aligned}
P(t) \quad &= \left(Z(t)^T Z(t)\right)^{-1} \\
&= \left([Z_{t-1}^T, Z^T(t)] \begin{bmatrix} Z_{t-1} \\ Z(t) \end{bmatrix}\right)^{-1} \\
&= \left(Z_{t-1}^T Z_{t-1} + Z^T(t) Z(t)\right)^{-1} \\
&= \left[P^{-1}(t-1) + Z^T(t) Z(t)\right]^{-1} \\
&= \left[I - \frac{P(t-1) Z^T(t) Z(t)}{1 + Z(t) P(t-1) Z^T(t)}\right] P(t-1)
\end{aligned}
\tag{9}
$$

c. The estimated value of the error parameter of each user's sub-meter is calculated as $\hat{\Theta}(t)$ using Eq. (10).

$$
\begin{aligned}
\hat{\Theta}(t) \quad &= \left(Z_t^T Z_t\right)^{-1} Z_t^T Y_t \\
&= P(t) [Z_{t-1}^T, Z^T(t)] \begin{bmatrix} Y_{t-1} \\ y(t) \end{bmatrix} \\
&= P(t) [Z_{t-1}^T Y_{t-1} + Z^T(t) y(t)] \\
&= P(t) [P^{-1}(t-1) \hat{\Theta}(t-1) + Z^T(t) y(t)] \\
&= P(t) \{[P^{-1}(t) - Z^T(t) Z(t)] \hat{\Theta}(t-1) + Z^T(t) y(t)\} \\
&= \hat{\Theta}(t-1) + P(t) Z^T(t) [y(t) - Z(t) \hat{\Theta}(t-1)]
\end{aligned}
\tag{10}
$$

d. The defined gain matrix is expressed as $K(t)$:

$$
K(t) = \frac{P(t-1) Z^T(t)}{1 + Z(t) P(t-1) Z^T(t)}
\tag{11}
$$

e. Combining Eqs. (8) with (11), when T $\leq$ L, the error check of the smart meter is given by Eq. (12).

$$
\begin{cases}
P(t) = P(t-1)[I - K(t-1) Z(t)] \\
\hat{\Theta}(t) = \hat{\Theta}(t-1) + K(t)[y(t) - Z(t)\hat{\Theta}(t-1)]
\end{cases}
\tag{12}
$$

Step 3: When $T > L$, the specific solution process of the LMRLSA module is as follows:

a. With the addition of a new set of data from the latest measurement period $T$, the calculation process of the inverse matrix $P(T - L, T)$ of the covariance of the measurement data is performed based on the measurement data of the $L + 1$ group of smart meters from the $(T$-L)th to the $T$-th and the recursive calculation results of the previous $T$-1 times:

$$
\begin{aligned}
P(T - L, T) \quad &= \left(Z(T - L, T)^T Z(T - L, T)\right)^{-1} \\
&= \left(Z(T - L, T - 1)^T Z(T - L, T - 1) + Z^T(T) Z(T)\right)^{-1} \\
&= \left[P^{-1}(T - L, T - 1) + Z^T(T) Z(T)\right]^{-1} \\
&= \left[I - \frac{P(T - L, T - 1) Z^T(T) Z(T)}{1 + Z(T) P(T - L, T - 1) Z^T(T)}\right] P(T - L, T - 1) \\
&= [I - K(T - L, T) Z(T)] P(T - L, T - 1)
\end{aligned}
\tag{13}
$$

In Eq. (13), the sub-meter reading matrix from $(T$-L)th to $T$ times is demonstrated in Eq. (14):

$$
\begin{aligned}
\boldsymbol{Z}(T-L,T) &= [\boldsymbol{Z}(T-L),\boldsymbol{Z}(T-L+1),\dots,\boldsymbol{Z}(T)]^T \\
&= \begin{bmatrix}
z_1(T-L) & z_2(T-L) & \cdots & z_m(T-L) \\
z_1(T-L+1) & z_2(T-L+1) & \cdots & z_m(T-L+1) \\
\vdots & \vdots & \ddots & \vdots \\
z_1(T) & z_2(T) & \cdots & z_m(T)
\end{bmatrix}
\end{aligned} \tag{14}
$$

b. $\boldsymbol{K}(T-L,T)$ is defined as the gain matrix, as illustrated in Eq. (15):

$$
\boldsymbol{K}(T-L,T) = \frac{\boldsymbol{P}(T-L,T-1)\boldsymbol{Z}^T(T)}{1+\boldsymbol{Z}(T)\boldsymbol{P}(T-L,T-1)\boldsymbol{Z}^T(T)} \tag{15}
$$

c. The estimated value of the error parameter of the meter is calculated as follows:

$$
\begin{aligned}
\hat{\Theta}(T-L,T) &= [\theta_1,\theta_2,\dots,\theta_m]^T \\
&= \left(\boldsymbol{Z}(T-L,T)^T\boldsymbol{Z}(T-L,T)\right)^{-1}\boldsymbol{Z}(T-L,T)^T\boldsymbol{Y}(T-L,T) \\
&= \boldsymbol{P}(T-L,T)\left[\boldsymbol{Z}(T-L,T-1)^T\boldsymbol{Y}(T-L,T-1)+\boldsymbol{Z}^T(T)y(T)\right] \\
&= \boldsymbol{P}(T-L,T)\left[\boldsymbol{P}(T-L,T-1)^{-1}\hat{\Theta}(T-L,T-1)+\boldsymbol{Z}^T(T)y(T)\right] \\
&= \hat{\Theta}(T-L,T-1)+\boldsymbol{K}(T-L,T)\times[y(T)-\boldsymbol{Z}(T)\hat{\Theta}(T-L,T-1)]
\end{aligned} \tag{16}
$$

Among them, the total meter reading matrix is expressed as $\boldsymbol{Y}(T-L,T) = [y(T-L),y(T-L+1),\dots,y(T)]^T$.

d. According to the above analysis, when a new set of T-th measurement data is added, the solution of the LMRLSA reads:

$$
\begin{cases}
\boldsymbol{K}(T-L,T) = \dfrac{\boldsymbol{P}(T-L,T-1)\boldsymbol{Z}^T(T)}{1+\boldsymbol{Z}(T)\boldsymbol{P}(T-L,T-1)\boldsymbol{Z}^T(T)} \\
\boldsymbol{P}(T-L,T) = [I-K(T-L,T)\boldsymbol{Z}(T)]\boldsymbol{P}(T-L,T-1) \\
\hat{\Theta}(T-L,T) = \hat{\Theta}(T-L,T-1)+\boldsymbol{K}(T-L,T)\times[y(T)-\boldsymbol{Z}(T)\hat{\Theta}(T-L,T-1)]
\end{cases} \tag{17}
$$

e. To maintain a constant memory length L unchanged, the measurement data from a group of smart meters at time $T$ is added, while the measurement data from time T-L needs to be removed. Based on the measurement data from the $(T-L+1)$th to the L sets of $T$-th and the calculation result obtained by Eq. (17), the LMRLSA can be obtained by excluding the measurement data from time T-L. The solution of the LMRLSA is defined as Eq. (18):

$$
\begin{cases}
\boldsymbol{K}(T-L+1,T) = \dfrac{\boldsymbol{P}(T-L,T)\boldsymbol{Z}^T(T-L)}{1-\boldsymbol{Z}(T-L)\boldsymbol{P}(T-L,T)\boldsymbol{Z}^T(T-L)} \\
\boldsymbol{P}(T-L+1,T) = [I+K(T-L+1,T)\boldsymbol{Z}(T-L)]\boldsymbol{P}(T-L,T) \\
\hat{\Theta}(T-L+1,T) = \hat{\Theta}(T-L,T)-\boldsymbol{K}(T-L+1,T)\times[y(T-L)-\boldsymbol{Z}(T-L)\hat{\Theta}(T-L,T)]
\end{cases} \tag{18}
$$

Step 4: Based on the actual data, the calculation example analysis is performed to conduct online verification and analysis of the estimation error of the electricity meter. This analysis includes the following contents:

a. The ratio between the total number of electricity meters and the sample size is determined.

b. Stratification is carried out according to the electricity consumption level of each user, thereby determining the number of smart meter samples drawn from each layer.

c. The sum of the number of smart meters drawn from each layer should be equal to the sample size.

d. For numbers that cannot be rounded, their approximate values are determined.

e. After extracting the actual values of the error parameters of the smart meter through on-site stratified sampling, the Mean Absolute Percent Error (MAPE) and the Root Mean Square Error (RMSE) are used as assessment criteria. In the process of remotely verifying the error of smart meters, a lower MAPE value and RMSE indicate a higher accuracy of the estimated error parameters (*Lv & Qiao, 2020*). If the total number of on-site samples is denoted as $p$, the MAPE and RMSE can be calculated using the following Eqs. (19) and (20):

$$MAPE = \frac{1}{p} \sum_{i=1}^{p} \frac{\left| \hat{\theta}_i - \theta_i \right|}{\theta_i} \times 100\% \tag{19}$$

$$RMSE = \sqrt{\frac{1}{p} \sum_{i=1}^{p} \left( \hat{\theta}_i - \theta_i \right)^2} \tag{20}$$

In Eqs. (19) and (20), $\theta_i$ means the actual error of the meter in the detected area, and $\hat{\theta}_i$ refers to the estimated value of $\theta_i$.

Subsequently, a hierarchical processing mechanism for abnormal situations in low-voltage areas can be developed. By incorporating the obtained results of the operation error of the smart meter, factors such as the magnitude of the operation error, its proportion within the batch, the underlying cause of the deviation, and the severity of the abnormality can be analyzed. This analysis enables the identification and replacement of faulty smart meters, as well as the provision of services such as online detection of electricity theft and electric leakage (*Feizi, 2020*; *Wan et al., 2021*).

## EXPERIMENTAL RESULTS AND ANALYSIS

### Verification of image recognition effect

The TensorFlow Object Detection API provides the corresponding eval.py script specifically for model verification. Executing the script can verify the test set and generate the corresponding log file. Then, the log file is accessed through TensorBoard, allowing for the visualization of the model's performance. Figures 7 and 8 display the graphical representation of the verification results, demonstrating the effectiveness of the model evaluation process.

The proposed step-by-step image recognition method is verified using a dedicated test dataset. The recognition accuracy is calculated based on the recognition results. The specific verification scheme is portrayed in Fig. 9.

The verification scheme uses the Remote Procedure Call (RPC) image recognition interface implemented within the system for verification. Then, the RPC interface facilitates

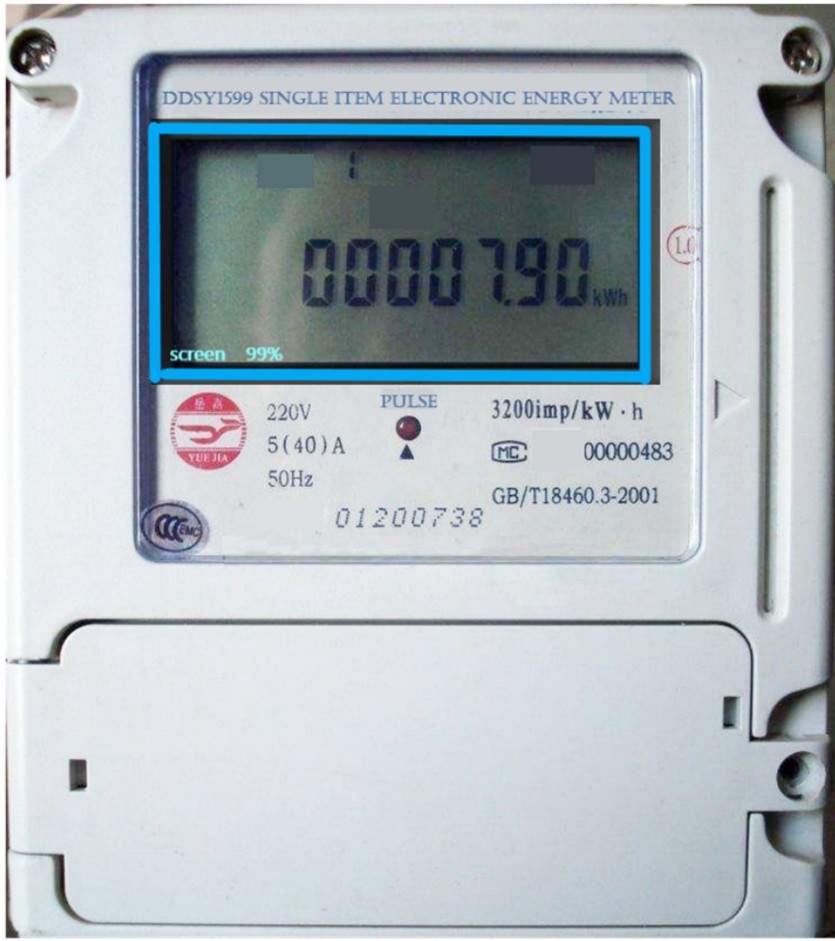

**Figure 7** **The SSD model detects the effective reading area of the electricity meter.** The TensorFlow Object Detection API provides the corresponding eval. py script to verify the trained model. Running the script can verify the test set and generate the corresponding log file. Then, the log file is read through TensorBoard and the verification effect of the model is displayed.

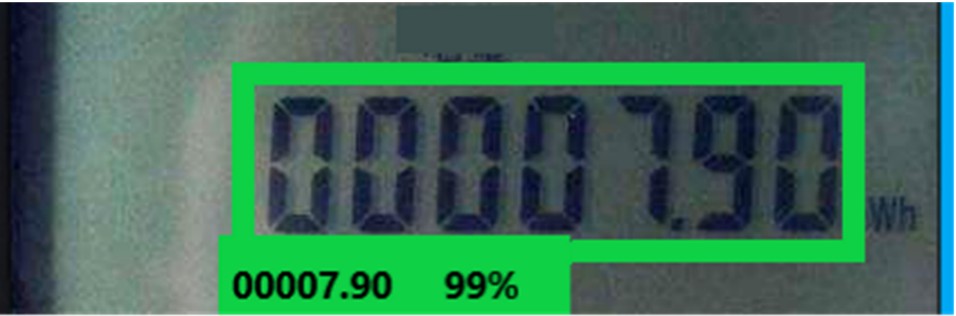

**Figure 8** **The Faster-RCNN model identifies the specific readings of the electricity meter.** The figure shows the specific display effect.

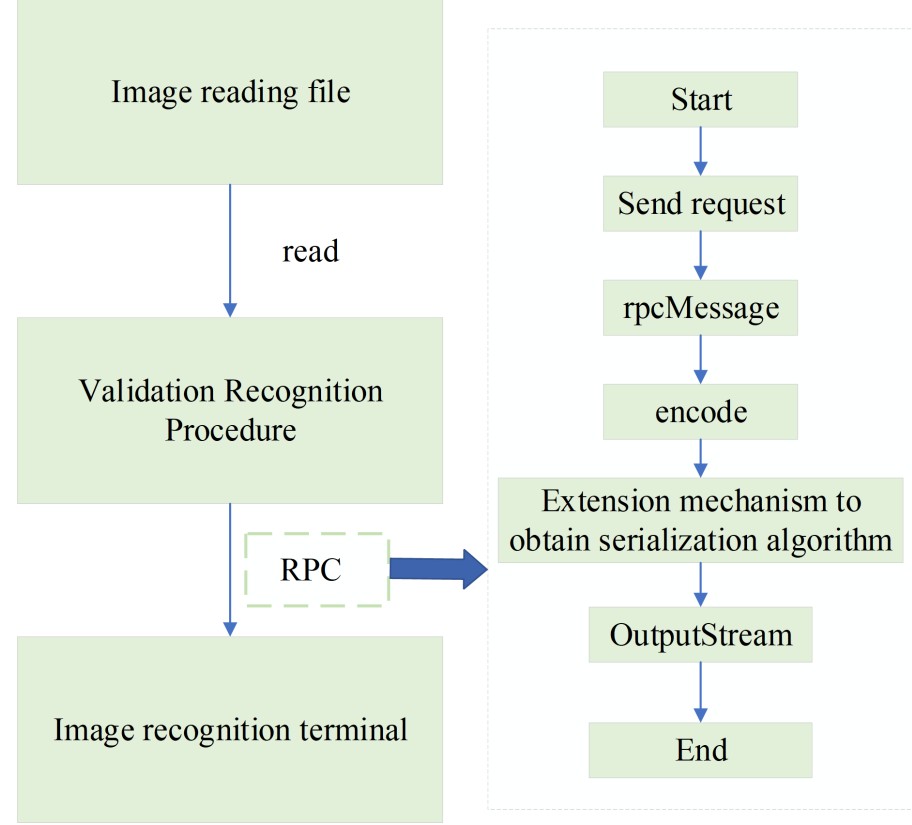

**Figure 9** **Verification scheme.** The proposed step-by-step image recognition method is verified using the test data set, and the accuracy of recognition is calculated according to the recognition results.

the invocation of the recognition service on the image recognition terminal, enabling the retrieval of recognition results. These results are then compared against the correct readings to assess the accuracy of the recognition system. A total of 528 electric meter images are used for testing. After conducting thorough testing and verification, the recognition accuracy is 98.49% when using the sample set of 528 images. Additionally, when using sample sets of 558 and 628 electric meter images, the recognition accuracies achieved 98.55% and 98.65%. Therefore, it is concluded that the proposed step-by-step image recognition method exhibits high effectiveness and accuracy in the recognition of electric meter readings.

## Validation of remote verification methods

To verify the effectiveness of the proposed method, this experiment utilizes real electricity meter measurement data collected from a city in China from February to May 2021, with a data collection frequency of 15min. The research area contains a total of 1 meter and 195 user sub-meters. Data pre-processing is conducted to filter measurements taken during no-load or light-load data conditions, resulting in the acquisition of distinct measurement data sets for analysis. The memory length L is assigned a value of 1,000. The resulting

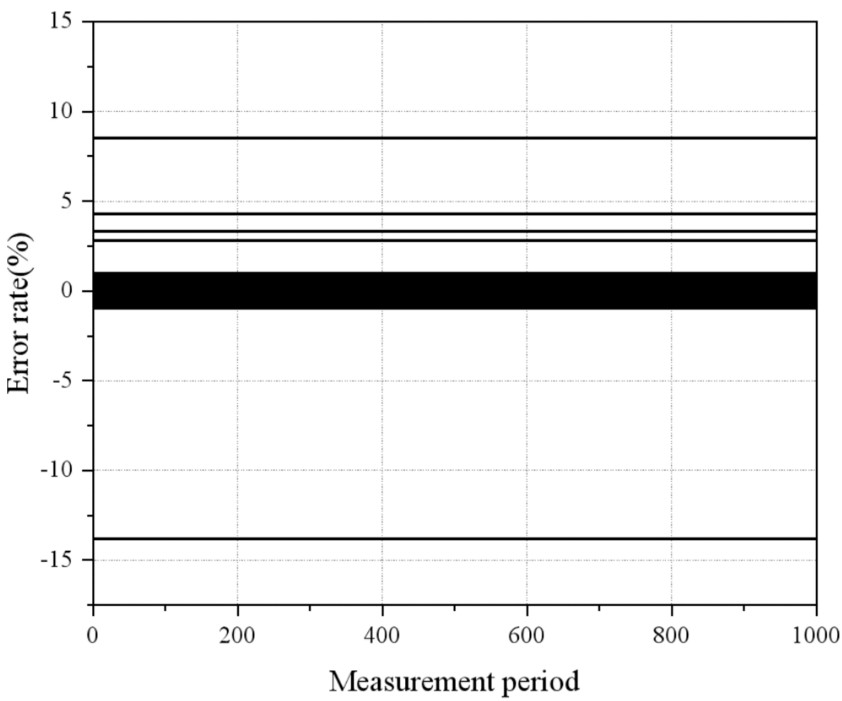

**Figure 10 The recursive estimation curve of the operation error of the electricity meter.** There are 5 m in the research area with extremely large errors, and the estimated error parameters of the remaining meters are all within the allowable range of normal errors.

recursive estimation curve of the operation error of the smart meter, obtained through the proposed method, is represented in Fig. 10.

Figure 10 illustrates that within the research area, there are 5 m with substantial errors, while the estimated error parameters of the remaining meters fall within the allowable range of normal errors. By considering data from a given measurement period, the estimated error value of the electricity meter for that specific period can be obtained, as detailed in Fig. 11.

Figure 11 indicates that most of the error rates among the selected user sub-meters in the designated power distribution area are within the allowable range of normal errors. However, user sub-meters No. 48, 65, 112, 135, and 181 exhibit errors that surpass the tolerance threshold. The error rates of the No. 48, 65, 112, 135, and 181 m are 4.2885%, 6.9741%, −14.1072%, 3.9381%, and 4.5189%, respectively. The user profile information obtained by the user information collection system enables accurate identification of the corresponding electricity customer information associated with the out-of-tolerance smart meters. A further collection of smart meter data allows for a detailed analysis and verification of the specific reasons behind the observed errors. For example, a positive error indicates potential aging and damage to the electricity meter, while a negative error suggests artificial tampering or electricity theft. Figure 11 reveals a prevalence of these two types of errors, indicating their common occurrence. Over time, internal components of the electricity meter may experience wear or decay, leading to increased measurement

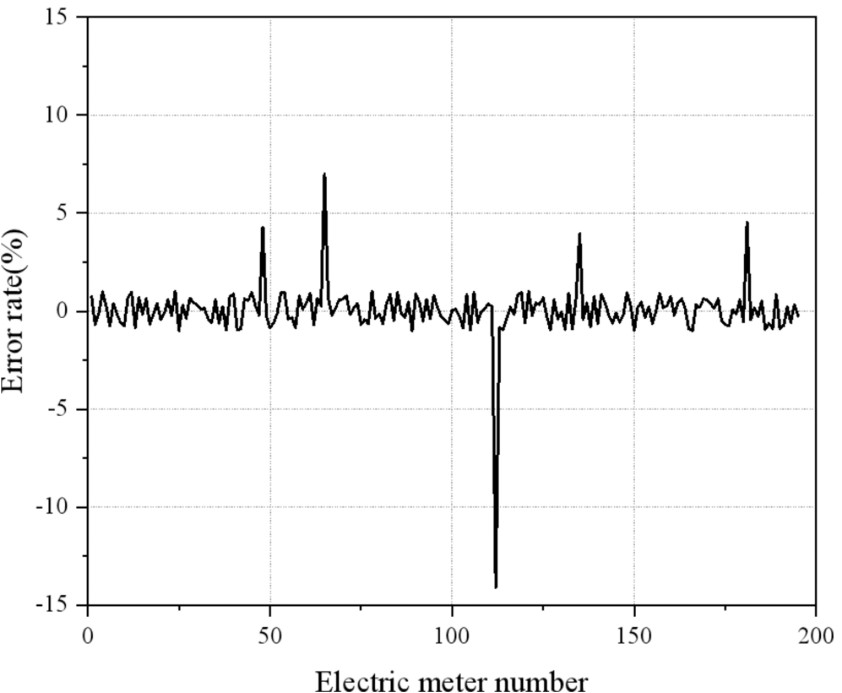

**Figure 11** **The estimation error value of the electricity meter in a certain period.** Most of the error rates of the user sub-meters in the selected power distribution area are within the allowable range of normal errors. User sub-meters No. 48, 65, 112, 135, and 181 have errors out of tolerance.

errors and deviations between meter readings and actual energy consumption. Moreover, burn-in effects can contribute to a decline in overall meter accuracy by approximately 5% to 10%, particularly under prolonged usage and high loads. Regular maintenance and calibration procedures are thus crucial for ensuring accuracy and reliability.

Based on the operational characteristics of the research area, an analysis is conducted to determine the optimal value range of the memory length L. The value of L in the proposed method is influenced by the frequency of the measurement data collected by the electricity meter in the area. Figure 12 illustrates the estimation error values of the electricity meter for different L values.

Figure 12 demonstrates the impact of different values of L on the estimation error of smart meters. It is observed that when L is set to 100, the estimation error significantly deviates from the actual error value of a normal smart meter. This discrepancy arises due to an insufficient number of recursive estimation equations, resulting in an underdetermined state in relation to the number of error parameters to be estimated. Consequently, the results cannot be applied. When L is increased to 400, the number of measurements surpasses the number of error parameters to be estimated, leading to convergence in the estimation error values. However, some normal smart meters still exhibit estimation error values that fall outside the acceptable tolerance range, indicating room for improvement in the estimation effect. By setting L to 1000, the estimation error values of each electricity meter approach a specific value, resulting in a more accurate estimation of the error

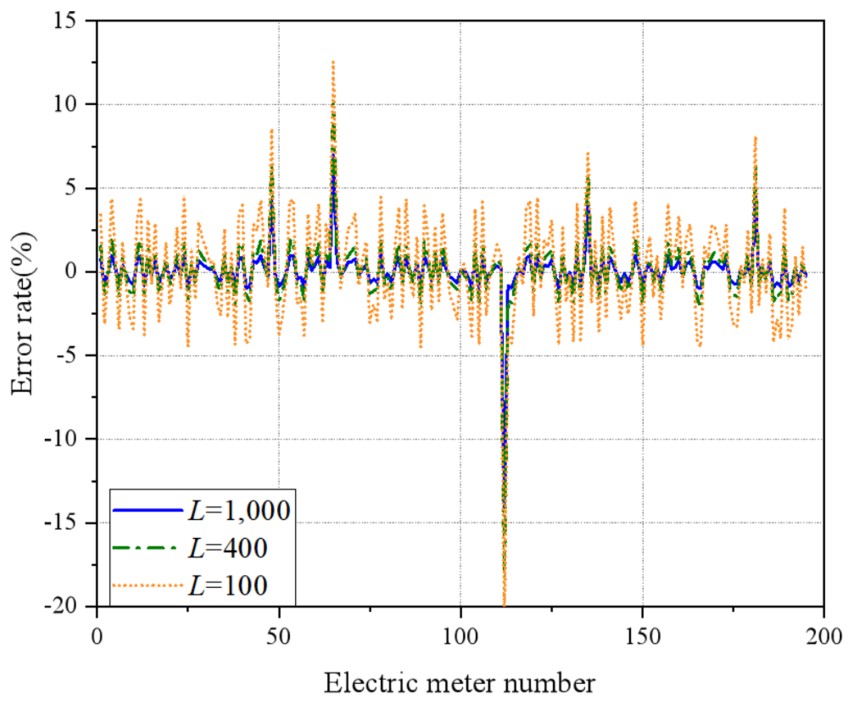

**Figure 12 The estimation error value of the electricity meter with different memory length.** When $L =$ 100, the estimation error value of a normal smart meter seriously deviates from its actual error value, because the number of recursive estimation equations is in an underdetermined state when it is less than the number of error parameters to be estimated. The results cannot be applied. When $L = 400$, the number of measurements is greater than the number of error parameters to be estimated, and the estimation error value begin to converge. However, the estimation error value of some normal smart meters are still in the out of tolerance range, and the estimation effect is not ideal. When $L = 1,000$, the estimation error value of each electricity meter is close to a certain value, and a more accurate estimation of the error parameters of the smart meter is obtained, and the estimation effect is ideal. When the value of L is large, although the error can also be estimated parameters, it takes a long time and reduces the efficiency of online analysis. Therefore, based on the actual working conditions of the research area, to ensure the accuracy of the estimated value and the real-time performance of the solution analysis, the recommended value of L ranges from 600 to 1,200.

parameters of the smart meter. This yields an ideal estimation effect. Nonetheless, when L is excessively large, although error parameters can still be estimated, it prolongs the analysis process and reduces the efficiency of online analysis. Therefore, considering the actual working conditions of the research area and aiming to ensure both accurate estimation results and real-time performance of the solution analysis, it is recommended to set the value of L within the range of 600 to 1,200.

In this research, the line loss is considered in the calibration of the electricity meter. Thus, significant calculation errors in the line loss can also impact the estimation result of the operation error of the electricity meter. Figure 13 provides a comparison of the estimation error value of each electricity meter under different calculation error rates of the line loss, namely 1%, 5%, 8%, and 10%.

Figure 13 shows that when the error in calculating the line loss is below 5%, it has minimal impact on identifying the out-of-tolerance electricity meter in the area. Accurate

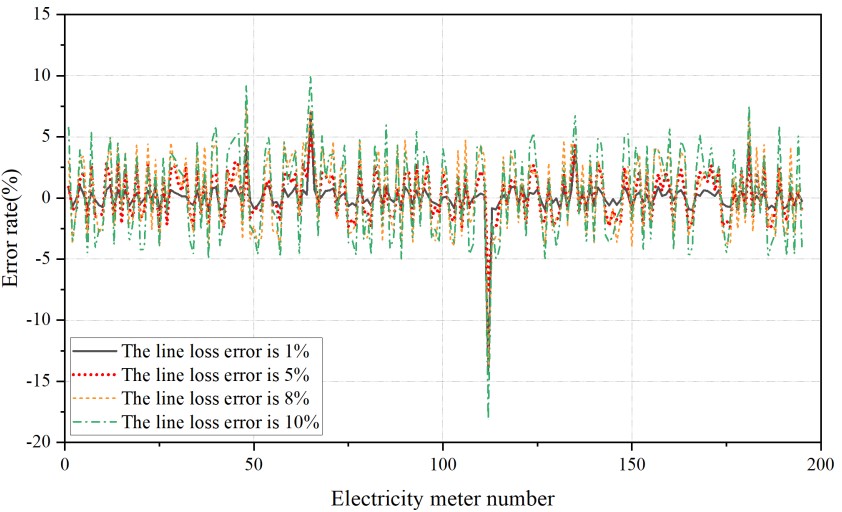

**Figure 13    The estimation error value of electricity meters under line losses of different errors.** When the error of the line loss calculation result is within 5%, it has little effect on finding the out-of-tolerance electricity meter in the area, and the out-of-tolerance electricity meter can be accurately found, and the effect is ideal. when the error of the line loss calculation result is about 8%, although the out-of-tolerance meter can be found, the estimation error of the normal electricity meter is higher than its actual error value. There is a situation of wrong detection, and the estimation effect is not good. When the error of the line loss calculation result is more than 10%, the obtained out-of-tolerance meter has the situation of missed detection and wrong detection, and the estimation effect is poor.

detection of out-of-tolerance meters is achieved, yielding satisfactory results. However, when the error in line loss calculation approaches 8%, exceeding their actual error values. This leads to instances of incorrect detection, indicating a need for improvement in estimation accuracy. Moreover, with line loss calculation errors exceeding 10%, there are cases of missed detection and false detection of out-of-tolerance meters, resulting in a poor estimation effect. The above analysis manifests the importance of accurate line loss to minimize its influence on the estimation of operating errors in electricity meters. The acceptable range of error is between 1% and 10%, and current technical methods necessitate a line loss calculation accuracy of less than 5%.

Data pre-processing is required to exclude data under light load conditions. In the analysis of the light load condition, the proposed algorithm is used to estimate the operation error of the smart meter using the measurement data obtained through clustering. The estimation results of the performance of the algorithm under light load conditions are manifested in Fig. 14.

Figure 14 illustrates the challenges encountered when estimating the operating error of electricity meters using measurement data under light load conditions. In such scenarios, it becomes difficult to accurately identify out-of-tolerance meters and a significant portion of the estimation error of the electricity meter fall within the out-of-tolerance range. This phenomenon can be attributed to two primary factors. Firstly, the irregular pulse generated by smart meters under light load conditions can cause creeping, further complicating the estimation process. Secondly, the increased proportion of excitation loss and iron loss in

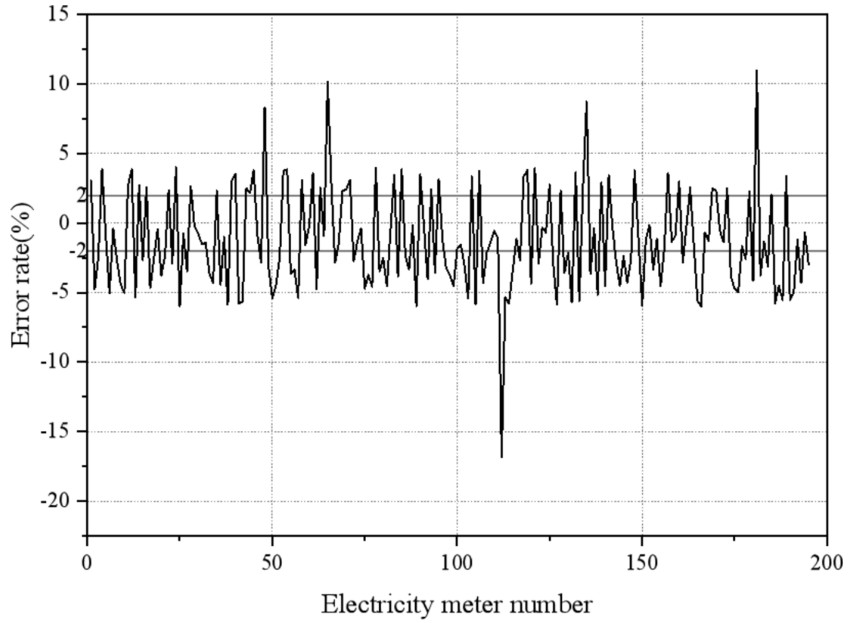

**Figure 14 Estimation error value of electricity meter without excluding light load condition.** When the measurement data under light load is used to remotely estimate the operating error of the electricity meter, it is impossible to determine the out-of-tolerance meter, and most of the estimation error of the electricity meter belong to the out-of-tolerance range.

the current transformer (CT) at light load affects the error characteristic of the CT, causing it to enter the nonlinear region. Consequently, there is no clear relationship between the error and the load, making accurate error compensation challenging. To ensure the accuracy of the estimation error for each user's smart meter and avoid over-checking, it is crucial to remove measurement data collected under light load conditions during the data pre-processing stage.

## CONCLUSION

The increasing scale of China's power grid necessitates the development of efficient and accurate methods for verifying electricity meters. The purpose is to simplify the meter verification method and enhance work efficiency without compromising calibration accuracy. An integrated approach is proposed to achieve this objective. Firstly, the automatic meter reading is realized through the Faster-RCNN and the SSD models, enabling the collection of metering information that is aggregated and transmitted to the master station. Secondly, the master station pre-processes the original measurement data, excluding data obtained under light load conditions. Thirdly, an estimation error model and solution equation of the electricity meter are established based on the pre-processed data. The operation error of the electricity meter is estimated, and the LMRLSA algorithm checks the accuracy of the estimation. Business assistant decision-making is then conducted based on the remote verification results. Furthermore, the accuracy of the image recognition system, employing the proposed Faster-RCNN and SSD models, is evaluated by testing the image

recognition effectiveness. Test results show recognition accuracies of 98.49%, 98.55%, and 98.65% when using 528, 558, and 628 electric meter images, respectively. Through remote verification testing, it is demonstrated that the proposed algorithm effectively mitigates the influence of outdated measurement data on error parameter estimation, improves the accuracy of the error parameter estimation, and enables real-time error estimation by adjusting the memory length. The recursive estimation curve facilitates online monitoring of electricity theft or leakage. Moreover, by analyzing the influence of parameters on error results, it is revealed that a memory length ranging from 600 to 1,200 and a line loss error of less than 5% yield the most suitable accuracy for electricity meter error estimation. It is recommended to remove measurement data collected under light load conditions to prevent over-checking. However, due to limited resources, the current solution requires further development in the backend to enhance data operations. Future improvements will focus on enhancing the operation experience. This research holds significant reference value for achieving intelligent remote online meter reading, verification, and management of electricity meters.

### Funding
The authors received no funding for this work.

### Competing Interests
The authors declare there are no competing interests. Peng Yang is employed by State Grid Hebei electric power company. Chong Li, Hao Wang, Hongtao Shen, Yi Wang, Qian Li, Chuan Li, Bing Li, Rongkun Guo, and Ruiming Wang are employed by State Grid Hebei Marketing Service Center.

### Author Contributions
- Chong Li conceived and designed the experiments, performed the computation work, authored or reviewed drafts of the article, and approved the final draft.
- Hao Wang conceived and designed the experiments, performed the computation work, prepared figures and/or tables, and approved the final draft.
- Hongtao Shen conceived and designed the experiments, performed the computation work, authored or reviewed drafts of the article, and approved the final draft.
- Peng Yang conceived and designed the experiments, prepared figures and/or tables, and approved the final draft.
- Yi Wang performed the experiments, performed the computation work, prepared figures and/or tables, and approved the final draft.
- Qian Li performed the experiments, prepared figures and/or tables, and approved the final draft.
- Chuan Li performed the experiments, performed the computation work, authored or reviewed drafts of the article, and approved the final draft.

- Bing Li analyzed the data, authored or reviewed drafts of the article, and approved the final draft.
- Rongkun Guo analyzed the data, prepared figures and/or tables, authored or reviewed drafts of the article, and approved the final draft.
- Ruiming Wang analyzed the data, authored or reviewed drafts of the article, and approved the final draft.

## Data Availability

The raw data are available in the Supplemental Files.

## Supplemental Information

Supplemental information for this article can be found online at http://dx.doi.org/10.7717/peerj-cs.1581#supplemental-information.

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
