# Peer review of "Online verification and management scheme of gateway meter flow in the power system by machine learning"

_PeerJ Computer Science, doi:10.7717/peerj-cs.1581_

## Round 0.1 · original submission · Minor Revisions

In regards to this paper, two reviewers have returned their comments and suggested a minor revision. In this case, I suggest a minor revision for further consideration.

·

Basic reporting

The paper aims to address the problem of verifying and managing the flow of gateway meters in the power system using multiple machine learning algorithms to continuously monitor flow in real-time. The novelty of the paper is the automation method of securing the integrity of gateway meter flow by focusing on a machine learning-based approach to enhance the power system process and improve accuracy.

All the figures have a strong relationship to the paper and experiment design. Each figure is well labeled & described. The paper will benefit greatly if following minor changes were done to the graphs:

1. Figure 3 & Figure 4: For the box “LabelImg Labeling”, it would be great if the authors could replace it with a more concise name.
2. Figure 6: The process of scheme; rescale all the boxes so they are the same size.
3. Figure 9: Verification Scheme; if possible, maybe expand a bit more on the validation procedure and RPC call.
4. Figure 13: the color of the second line(the line loss error is 5%) is a bit hard to distinguish from the other 3, I’d suggest changing the color to a more vibrant one such as red.

Raw data is supplied. Source code includes drawing the plots but the portion with machine learning models(e.g. Pre-trained model mentioned in the paper using TensorFlow) is not provided.

Few things with the source code include:
1. There are a few lines containing unidentified symbols at the beginning of each row. Please make sure you are using ASCII values in C++ vs UTF-8.
2. Please make sure each function is correctly formatted and anything that’s not code is commented out.

References are all complete.

Experimental design

The experiment's design is quite straightforward. The automatic meter reading is done using Faster-RCNN and SSD models and data are further processed to measure the accuracy. The baseline is obviously set to be the original verification and management process in power systems.
For line 110:”...a training sample set is constructed…”, is the whole labeled data used for training samples?

For line 363 - 364: ‘... it is verified that the image recognition system based on the proposed Faster-RCNN model and the SSD model has an accuracy rate of 98.49%...“ and in line 277 “...528 electric meter images are used for testing...”, the testing data is not large enough for the model accuracy to be considered a generalized approach. If the situation allows, it would be great for authors to include more details on experiment comparisons. For example, accuracy when different testing sizes(or more testing data can be used?). And how robust and generalized the proposed process like for meters/power systems in different cities/regions.

For the Line 304: “...destroyed, and there is electricity stealing behavior…”. Is there any data or statistics on how prevalent this situation is and how much it affects the final accuracy?

Validity of the findings

The findings are reached by running pretrained machine learning models with real-world data on devices. The premises of the experiment design and the limitation with the current gateway meter flow in power systems are identified by authors via domain knowledge and real-world examples. In summary the manuscript proposed an efficient way to improve the existing process for gateway meter flow in the power systems. The accuracy improvement seems sound and there are no bold claims. The findings presented in this paper can be considered valid.

Additional comments

Thank you for giving me the opportunity to review this manuscript. The paper is well written and well-organized. It is always nice to see machine learning models in the applied fields and the fact that the algorithm could offer such insights on real-world data could greatly help researchers and engineers gain insights on flow monitoring and control. The experiment's design has a strong relation to the model and the problem and no claims in the paper were purposely done to reach incorrect conclusions. Overall, the study and findings presented in this paper holds promise in advancing the field. For the purposes of this journal I think it can be published after addressing the comments above.

Please note that this review is based on the information provided in the manuscript. If any relevant details were omitted, please consider including them in the revised version.

·

Basic reporting

1. References for figures used in the article (if any)
2. Extensive description about the exiting research is missing from the article

Experimental design

1. As per article there are about 500 million meters across China, are 500 images used for experiments, is enough data for the experiments ?

2. If Faster-CNN is running slow, are there any other algorithms (other than SDD) which can do the same job faster ? OR can same algorithm be used by breaking the data into smaller subsets ? OR can parameters be tunned for the same algorithm ?


3. Does images were input one by one OR all 500 together to train the model ?

Validity of the findings

1. If SDD is faster , why not use SDD only ? Why combination of F-CNN and SDD ?

2. How are results differ for F-CNN and SDD ?

3. What is the allow range of error ? What is the out of range error ? is there any numerical value ?

4. What different information does "Estimated Error value of Electricity meter without excluding light rod condition " and "Estimated Error value of Electricity meter under losses of different error" explaining ?

Additional comments

1. Article is well written but missing some of the existing research,

2. Article is little more around the mathematical equations and formulas for the analysis instead of Literature Review and findings

---

## Round 0.2 · accepted · Accept

As per comments from both original reviewers, this revised paper can be accepted for publication now.

·

Basic reporting

The revised paper provided more data and experiment details on the existing online remote verification methods for smart meters and it’s drawback analysis. In a addition, there are more details on the design of the Faster-RCNN model and SSD model, which further proves the novelty of the paper, using the automation method of securing the integrity of gateway meter flow by focusing on a machine learning-based approach to enhance the power system process and improve accuracy.

The updated experiment flow chart as well as model architecture graph is supplied.

All references are complete.

All code and source data are complete.

Experimental design

The experiment's design is easy to understand. The automatic meter reading is done using Faster-RCNN and SSD models and data are further processed to measure the accuracy. The revised paper did answer the questions raised in the first version of paper review including training data, accuracy review and how robust and generalized the proposed process like for meters/power systems in different cities/regions.

Validity of the findings

The findings reached by experiments design and analysis done in this paper are valid. The results are done by running pretrained machine learning models with real-world data on devices and the situation is specific to regions and infrastructure setups. The accuracy improvement seems sound and there are no bold claims. The findings presented in this paper can be considered valid.

·

Basic reporting

Based on comments from previous review, the author has answer all the question thoroughly and covered the missing gap between the Literature review.

One more is good to have to explaining the analysis and measurement data

Experimental design

Based on comments from previous review, the author has answer all the question thoroughly

Please mention about how images were input to train the model (if not)

Validity of the findings

Based on comments from previous review, the author has answer all the question thoroughly


Please mention about Estimated Error value of Electricity meter without excluding light rod condition " and "Estimated Error value of Electricity meter under losses of different error (if not)

Additional comments

Article is written in well English and shows the problem really well.